# An Improved Acoustic Pick-Up for Straight Line-Type Sagnac Fiber Optic Acoustic Sensing System

**DOI:** 10.3390/s22218193

**Published:** 2022-10-26

**Authors:** Jianjun Chen, Jiang Wang, Ning Wang, Juan Ruan, Jie Zhang, Yong Zhu

**Affiliations:** 1The Key Laboratory of Electromagnetic Technology and Engineering, Nanchong of Sichuan, China West Normal University, Nanchong 637000, China; 2The Institute of Xi’an Aerospace Solid Propulsion Technology, Xi’an 710025, China; 3The Key Laboratory of Optoelectronic Technology & System, Education Ministry of China, Chongqing University, Chongqing 400044, China

**Keywords:** acoustic detection, pick-up, Sagnac, optic fiber sensing

## Abstract

An improved acoustic pick-up is presented to enhance the acoustic sensing sensitivity of the straight line-type Sagnac fiber optic acoustic sensing system. A hollow elastomer cylinder is introduced into the system, and the optical fiber is tightly wound around the cylinder to construct the pick-up. The theoretical analysis was finished, and it showed that the improved pick-up will bring in an extra phase change and let the phase difference increase almost an order. The extra phase change will enhance the sensing sensitivity correspondingly. The titanium alloy elastic cylinder was designed and manufactured. The experiment system was built, and a sinusoidal acoustic signal was used as the sound source. The tests were taken during 100–1500 Hz, and the experimental study showed that the sensitivity was more than 130 mV/Pa when the pick-up was used as the acoustic sensing element, and compared to the fiber only system, the sensitivity was enhanced more than 71.2%. The improved pick-up will be helpful in sound recognition and expand the application area of the Sagnac acoustic sensing system.

## 1. Introduction

With the maturity of fiber optic sensing technology, the application field of optical fiber sensing continues to expand. Fiber optic acoustic sensing has been widely studied and applied in the fields of intrusion detection, environmental noise monitoring and other fields because of its excellent environmental tolerance, such as anti-electromagnetic interference and anticorrosion [1,2,3,4]. Among much research on acoustic sensing, the acoustic sensing system based on Sagnac interference is favored by researchers, because it has high sensitivity, a high signal-to-noise rate and, especially, can be applied in harsh environments [5,6,7].

A straight line-type Sagnac fiber optic acoustic sensing system is proposed for a no man’s border to monitor the noise coming from the military motor vehicles, armored cars, unmanned aerial vehicles, etc. in a wild environment. The system significantly reduces the length of the Sagnac interference fiber ring and avoids the exposure of a large number of non-sensing fibers to the working environment. The environmental adaptability of the system is significantly improved, and the application field is broadened. However, the sound sensitivity becomes the main factor that restricts the wide application of the straight line-type Sagnac fiber optic acoustic sensing system [8,9,10]. Researching the structure of acoustic fiber optic pick-up and improving the sound response sensitivity have become the focus of the straight line-type Sagnac fiber optic acoustic sensing system [11,12,13].

In this paper, an improved acoustic pick-up was designed, and the theoretical analysis was finished. The contrast experiments between the optical fiber ring and titanium alloy pick-up were finished with a sinusoidal signal. The experimental results showed that the introduction of acoustic pick-up will be helpful to further research on sound recognition.

## 2. Pick-Up Configuration and Theory Analysis

As shown in Figure 1, the straight line-type Sagnac fiber optic acoustic sensing system shorts the length of the non-sensing optic fiber. The light from the SLD (super-luminescent diode) is divided into three beams by coupler C_1_, and two beams of them enter the CW (clockwise) and CCW (counterclockwise) optical paths. Then, the two beams are merged into one single beam and transmitted to the optical fiber acoustic pick-up ring L_2_. At coupler C_3_, the light is divided into two beams and comes back through ports 2 and 3 of C_3_. When the two beams come back to C_1_, interference occurs and will be detected by a PD (photoelectric detector). In the system, ports 2 and 4 of C1 were not used. Compared with the traditional Sagnac optical fiber acoustic sensing system, the non-sensing optic fiber is shortened nearly 50%, and as shown in the dashed box in Figure 1, the components exposed to the field environment are significantly reduced. The environmental adaptability of the system is significantly improved.

As shown in Figure 1, the sound pick-up only relies on the optical fiber ring, and the sensitivity of the system is very low, so the collection, analysis, and recognition of acoustic signals hardly happen. Therefore, introducing a suitable pick-up structure to amplify the effect of sound on the optical fiber is significant for the system.

As shown in Figure 2, a hollow elastomer cylinder is introduced into the system as a sound-sensitive element. The sensing fiber is tightly wound around the outer wall of the elastic cylinder. When acoustic pressure acts on the elastic cylinder, the length, core diameter, and core refractive index of the sensing fiber will change. Then, the phase of the light is changed, and the interference will occur, and the sound can be detected from the interference light.

When an optical fiber ring is placed in a sound field, the phase change caused by the acoustic pressure is
(1)$$\Delta \varphi =\beta \Delta L+L\Delta \beta =\beta L\frac{\Delta L}{L}+L\frac{\partial \beta }{\partial d}\Delta d+L\frac{\partial \beta }{\partial n}\Delta n$$
where *φ* is the phase delay, *n* is the core refractive index, *L* is the length of fiber ring, and *β* is the propagation constant. *d* is the diameter of the fiber core. The first term of Formula (1) is the strain effect, which is caused by the fiber length change, the second term is the Poisson effect, which is caused by the fiber diameter change, and the third term is the elasto-optical effect, which is caused by the refractive index change. Usually, the diameter change of the fiber core is minuscule, and the second term is negligible. The phase change caused by the sound field can be simplified as
(2)$$\Delta \varphi =\beta \Delta L+L\Delta \beta =\beta L\frac{\Delta L}{L}+L\frac{\partial \beta }{\partial n}\Delta n=\Delta \varphi_{{}_{L}}+\Delta \varphi_{{}_{n}}$$

As shown in Figure 3, the optical fiber is the isotropic medium, the shear stress can be ignored, and the deformation quantity is related only to normal stress. The whole phase change caused by the sound field can be received as [5,14,15,16].
(3)$$\begin{array}{cc} \Delta \varphi & =\Delta \varphi_{L}+\Delta \varphi_{n}\\ & =\frac{2\beta L\mu_{f}}{E_{f}}P+\frac{n^{2}L\beta \left[(1-\mu_{f})p_{11}+(1-3\mu_{f})p_{12}\right]}{2E_{f}}P\\ & =\frac{L\beta }{E_{f}}\left\{2\mu_{f}+\frac{n^{2}}{2}\left[(1-\mu_{f})p_{11}+(1-3\mu_{f})p_{12}\right]\right\}P \end{array}$$
where *E_f_* is the elasticity modulus of the optical fiber, *μ_f_* is the Poisson ratio of the optical fiber, *p*_11_ and *p*_12_ are the elastic-optic coefficient tensors, *p*_11_ is decided by linear stress and linear strain, *p*_12_ is decided by torsional stress and torsional strain, *n* is the initial refractive index, and *P* is the sound pressure. Let *k*_1_
*=* {2*μ_f_ + n^2^*[(1 − *μ_f_*)*p*_11_
*+* (1 − 3*μ_f_*)*p*_12_]/2}/*E_f_*, Equation (3) can be simplified as
(4)$$\Delta \varphi =\beta Lk_{1}P$$

When the optical fiber is tightly wound around the outer wall of the elastic cylinder, the phase change of the optical fiber not only comes from the effect of the sound field but can also be influenced by the elastic cylinder. When the elastic cylinder is put in the sound field, it can be approximated that the force from the sound field is uniform. The equilibrium state equation of the elastic cylinder is
(5)$$\frac{\partial \sigma_{r}}{\partial r}+\frac{\sigma_{r}-\sigma_{\theta }}{r}=0$$
where *σ_r_* is radial stress, *σ_θ_* is tangential stress, and *r* is radius. Since the load of the elastic cylinder comes from the optical fiber ring, the load has axial symmetry, and the axial strain is independent of *r* and *θ*. Therefore, the stress of the elastic cylinder can be expressed as
(6)$$\left\{ \begin{array}{l} \sigma_{r}=\frac{E}{1+\mu }(\frac{K_{1}+\mu K_{3}}{1-2\mu }-\frac{K_{2}}{r^{2}})\\ \sigma_{\theta }=\frac{E}{1+\mu }(\frac{K_{1}+\mu K_{3}}{1-2\mu }+\frac{K_{2}}{r^{2}})\\ \sigma_{z}=\frac{E\left[2\mu K_{1}+K_{3}(1-\mu )\right]}{\left(1+\mu )(1-2\mu \right)} \end{array}\right.$$
where *E* is the elasticity modulus; *μ* is the Poisson ratio from the elastic cylinder; *σ_z_* is the axial stress; and *K*_1_, *K*_2_, and *K*_3_ are constants.

As shown in Figure 4, the cylinder wall is very thin, the axial deformation is much less than the radial deformation, and the sound presses of the upper and lower end faces are ignored. The boundary conditions of the elastic cylinder can be expressed as
(7)$$\left\{ \begin{array}{l} \sigma_{r|r=a}=-P\\ \sigma_{r|r=b}=-(\sigma_{R}+P)\\ \sigma_{z}=\sigma_{\theta }=0 \end{array}\right.$$
where *K*_1_, *K*_2_, and *K*_3_ are from Equations (6) and (7).
(8)$$\left\{ \begin{array}{l} K_{1}=-\frac{(1-\mu )(1-2\mu )(\mu -1)b^{2}}{(2\mu^{2}+\mu -1)(b^{2}-a^{2})E}(\sigma_{R}+\frac{b^{2}+a^{2}}{b^{2}}P)\\ K_{2}=-\frac{(1+\mu )a^{2}}{(b^{2}-a^{2})E}\left[\sigma_{R}-\frac{\left(b^{2}-a^{2})(b^{2}-1\right)}{b^{2}}P\right]\\ K_{3}=-\frac{2\mu (1-\mu )(1-2\mu )b^{2}}{(2\mu^{2}+\mu -1)(b^{2}-a^{2})E}(\sigma_{R}+\frac{b^{2}+a^{2}}{b^{2}}P) \end{array}\right.$$

The counteracting force from the optical fiber ring can be expressed as
(9)$$\sigma_{R}=\frac{E_{f}S_{f}N\Delta b}{b^{2}H}=\frac{k_{f}N}{bH}\frac{\Delta r}{b}$$
where *k_f_* is the normalized elastic coefficient and *k_f_* = *E_f_S_f_*, *S_f_* is the cross-sectional area of the optical fiber, *N* is the optical fiber ring number of twisting, *H* is the effective height of the optical fiber ring, and Δ*r* is the radial displacement of the elastic cylinder. According to the Lame equation, Δ*r* can be expressed as
(10)$$\begin{array}{cc} \Delta r & =\Delta S_{r=b}\\ & =K_{1}b+\frac{K_{2}}{b}\\ & =-\frac{(1+\mu )(1-2\mu )(\mu -1)b^{2}}{(2\mu^{2}+\mu -1)(b^{2}-a^{2})E}(\sigma_{R}+\frac{b^{2}+a^{2}}{b^{2}}P)\\ & -\frac{(1+\mu )a^{2}b}{(b^{2}-a^{2})E}\left[\sigma_{R}-\frac{\left(b^{2}-a^{2})(b^{2}-1\right)}{b^{2}}P\right] \end{array}$$

Joining Equations (9) and (10), the relationship between Δ*r* and sound pressure *P* is
(11)$$\Delta r=-\frac{\left[m_{1}(b^{2}-a^{2})(1-b^{2})a^{2}b^{2}+m_{2}(a^{2}+b^{2})\right]d_{f}}{Em_{3}(b^{2}-a^{2})b^{3}d_{f}+\left[m_{4}a^{2}b^{2}+m_{2}\right]k_{f}}P$$

In Equation (11), *m*_1_, *m*_2_, *m*_3_, *m*_4_, *N*, and *H* can be expressed as follows:(12)$$\left\{ \begin{array}{l} m_{1}=(2\mu^{2}+\mu -1)(1+\mu )\\ m_{2}=(1+\mu )(1-2\mu )(\mu -1)\\ m_{3}=2\mu^{2}+\mu -1\\ m_{4}=(1+\mu )(2\mu^{2}+\mu -1)\\ N=\frac{L}{2\pi b}\\ H=Nd_{f}=\frac{Ld_{f}}{2\pi b} \end{array}\right.$$

Let $k_{2}=-\frac{\left[m_{1}(b^{2}-a^{2})(1-b^{2})a^{2}b^{2}+m_{2}(a^{2}+b^{2})\right]d_{f}}{Em_{3}(b^{2}-a^{2})b^{3}d_{f}+\left[m_{4}a^{2}b^{2}+m_{2}\right]k_{f}}$, and while the structure of the pick-up is determined, *k*_2_ is constant. Then, the optical propagation phase change caused by the deformation of the pick-up can be expressed as
(13)$$\Delta \varphi_{s}=\beta \Delta L_{s}=\beta L\frac{\Delta L_{s}}{L}=\beta L\frac{2\pi N\alpha \Delta r}{N2\pi b}=\beta L\frac{\alpha k_{2}P}{b}=\beta Lk_{3}P$$
where *k*_3_
*= αk*_2_*/b*, *α* is the deformation transfer coefficient, which is an empirical value. After introducing the pick-up structure, the whole phase change of the optical fiber ring can be expressed as
(14)$$\begin{array}{cc} \Delta \varphi_{P} & =\Delta \varphi_{L}+\Delta \varphi_{n}+\Delta \varphi_{s}\\ & =\frac{2\beta L\mu_{f}}{E_{f}}P+\frac{n^{2}L\beta \left[(1-\mu_{f})p_{11}+(1-3\mu_{f})p_{12}\right]}{2E_{f}}P+\beta Lk_{3}P\\ & =L\beta (k_{1}+k_{3})P \end{array}$$

An additional phase change appeared. When the single-mode silica fiber is used as the sensing fiber, *E_f_* = 7 × 10^10^ Pa, *d_f_* = 125 μm, *S_f_* = 1.23 × 10^−8^ m^2^, *n* = 1.456, *p*_11_ = 0.121, *p*_12_ = 0.270, *u_f_* = 0.1, *α* = 0.4, and for titanium alloy pick-up, *E* = 1.05 × 10 + 11 Pa, *μ* = 0.37, set *b* = 5.5 × 10^−2^ m, *a* = 5.2 × 10^−2^ m. It is easy to get *k*_1_ = 7.37 × 10^−10^ Pa^−1^, *k*_3_ = 6.092 × 10^−9^ Pa^−1^, while *k*_1_ + *k*_3_ = 6.829 × 10^−9^ Pa^−1^. Compared to a single fiber ring, the introduction of the pick-up structure makes the light propagation phase change in the sound field increase almost an order. Then, the sensitivity increases about 73.4% when the sensing fiber length set as 20 m, the light wavelength is 1310 nm, and the sound pressure is 1 Pa.

## 3. Results

Based on the calculations, a titanium alloy elastic cylinder was designed and manufactured. The titanium alloy pick-up is shown in Figure 5. The left part of Figure 5 is the pick-up. In order to enhance the stability of the experimental setup, a protection cover was designed. The pick-up was pasted to the pedestal, and the cover was pasted to the pedestal to protect the sensing fiber. 

As shown in Figure 5, the optical fiber ring is tightly wound on the upper part of the titanium alloy pick-up, and the experiment settings based on the Sagnac photoacoustic sensing system are shown in Figure 6.

Since the monitoring object of the system is focused on military motor vehicles, armored cars, and unmanned aerial vehicles, the detection frequency is mainly distributed as 500–1000 Hz. The contrast experiments between the optical fiber ring and titanium alloy pick-up were completed at 500 Hz and 1000 Hz using a sinusoidal signal. In the tests, the SLD named GR1346Q-A from the 44th Research Institute of CETC was used as the light source, and it has a central wavelength of 1310 nm, a spectral bandwidth more than 40 nm, and an output light intensity larger than 1 mW. An InGaAs detector named GT322D from the 44th Research Institute of CETC was used as the photoelectric detector; it has a spectral response range of 900–1700 nm and a spectral responsivity of 0.7 A/W at 1310 nm; the sensing fiber was single-mode and single-core silica fiber, which is bend-insensitive, the DAQ is LTC2297 from ADI, and the FPGA chip is XC6SLX45CSG324 from XILINX. The sound intensity variation ranges from 50 dB (0.0063 Pa) to 90 dB (0.63 Pa). The test results of the 500-Hz sound signal are shown in Figure 7.

Figure 7 shows that, when the optical fiber ring was replaced by pick-up, the sensitivity of the sensing system obviously improved. The fitted curve of the pick-up data is *V_Pout_* = 219.8*s + 70, and the sensitivity recaches 219.8 mV/Pa, while the fitted curve of the optical fiber ring is *V_Rout_* = 128.4*s + 69, and the sensitivity is 128.4 mV/Pa. It means, when the pick-up was used, the sensitivity increased by 71.2%.

The test results of 1000 Hz are shown as Figure 8. The fitted curve of the pick-up data is *V_Pout_* = 136.9*s + 68.4, and the sensitivity recaches 136.9 mV/Pa, while the fitted curve of the optical fiber ring is *V_Rout_* = 79.6*s + 68, and the sensitivity is 79.6 mV/Pa. It means that, when the pick-up was used, the sensitivity increased by 71.9%.

For further research on the sound frequency spectral response of the system with titanium alloy pick-up, the frequency test was finished. In the experiment, the sound frequency changed between 100 and 1500 Hz.

As shown in Figure 9, the highest sensitivity appeared at 5000 Hz, and when the frequency moves away, the sensitivity will gradually decrease. During 100–1500 Hz, the sensitivity is always more than 130 mV/Pa.

## 4. Discussion

This paper introduces an elastic metal cylinder pick-up instead of the fiber ring in the straight line-type Sagnac fiber optic acoustic sensing system to enhance the sensitivity. The elastic cylinder is made of titanium alloy material, and the sensing fiber ring is tightly wound around the outer wall of the titanium alloy cylinder to constitute the pick-up. The theoretical model was established, and the phase change came from the theoretical calculations. The calculation results show that the phase change has an extra increment because of the pick-up. The pick-up based on the titanium alloy elastic cylinder was produced, and the experimental tests were finished. Tests results show that the sensitivity of the system is improved from 128.4 mV/Pa to 219.8 mV/Pa for the 500-Hz acoustic signal, and the sensitivity is improved from 79.6 mV/Pa to 136.9.3 mV/Pa for the 1000-Hz acoustic signal. Overall, the introduction of the pick-up enhanced the sensitivity more than 71.2% compared to the system only using the fiber ring. The experimental results show that the introduction of the pick-up structure can significantly improve the sensitivity of the system. Meanwhile, the sound frequency spectral tested results show that the sensitivity is more than 130 mV/Pa during 100–1500 Hz, which can be used for sound source location and the recognition of military motor vehicles, armored cars, unmanned aerial vehicles, and so on in the wild environment.

## 5. Conclusions

In summary, a sound pick-up for the straight line-type Sagnac fiber optic acoustic sensing system was presented, which significantly improved the sensitivity. The key conclusions are as follows:(1)The hollow elastomer cylinder pick-up was designed, and the theoretical analyses showed that an extra phase change was gotten because of the pick-up structure, the phase difference of the system was improved by almost an order, and the sensitivity of the system was improved significantly.(2)The pick-up was produced with a titanium alloy elastic cylinder, and the experiments were finished. The results showed that the sensitivity was more than 130 mV/Pa during 100–1500 Hz, and the sensitivity was improved more than 71.2% compared with the sensing system only using the sensing fiber.(3)The introduction of the pick-up can significantly improve the sensitivity of the straight line-type Sagnac fiber optic acoustic sensing system, which will make the sound detection and identification become easier.

This work solved the low sensitivity problem and expanded the application fields of the straight line-type Sagnac fiber optic acoustic sensing system.

## Figures and Tables

**Figure 1 sensors-22-08193-f001:**
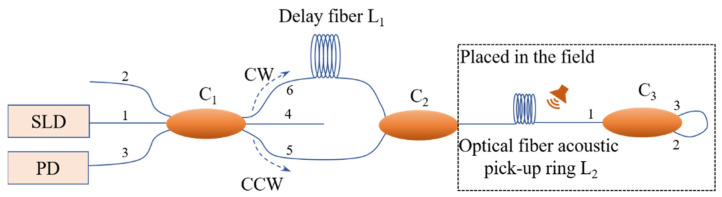
The schematic of the straight line-type Sagnac fiber optic acoustic sensing system.

**Figure 2 sensors-22-08193-f002:**
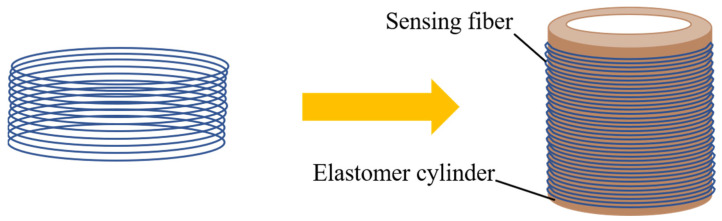
The schematic diagram of the straight line-type Sagnac fiber optic acoustic sensor.

**Figure 3 sensors-22-08193-f003:**
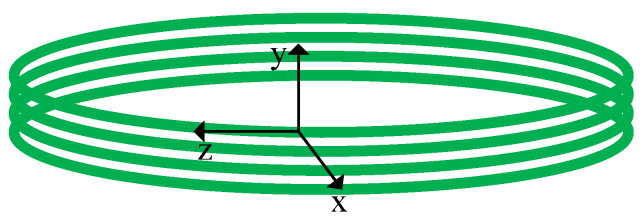
Schematic diagram of the fiber stress coordinates.

**Figure 4 sensors-22-08193-f004:**
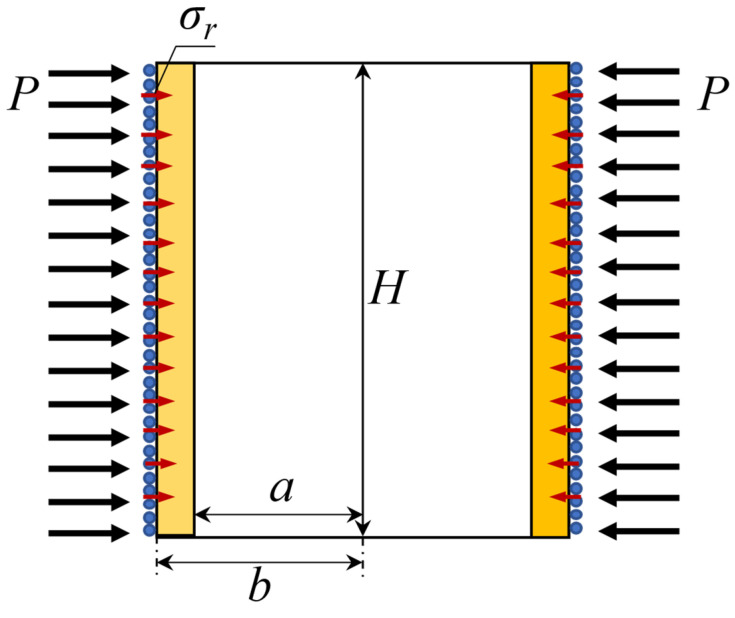
Pick-up boundary condition analysis diagram.

**Figure 5 sensors-22-08193-f005:**
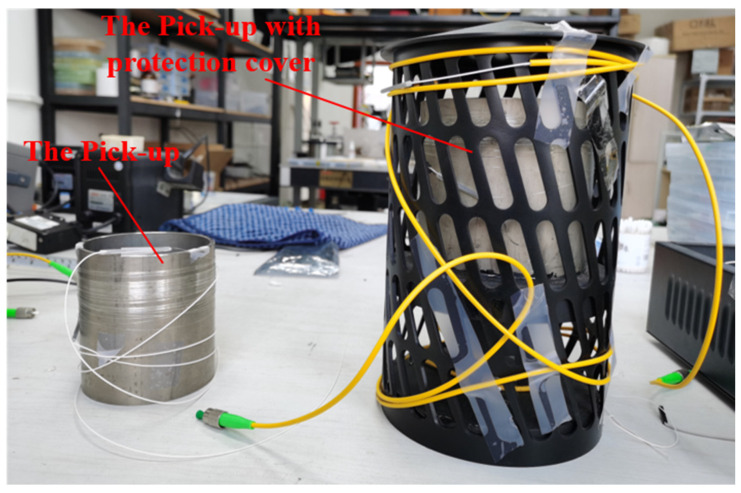
The photo of a titanium alloy pick-up.

**Figure 6 sensors-22-08193-f006:**
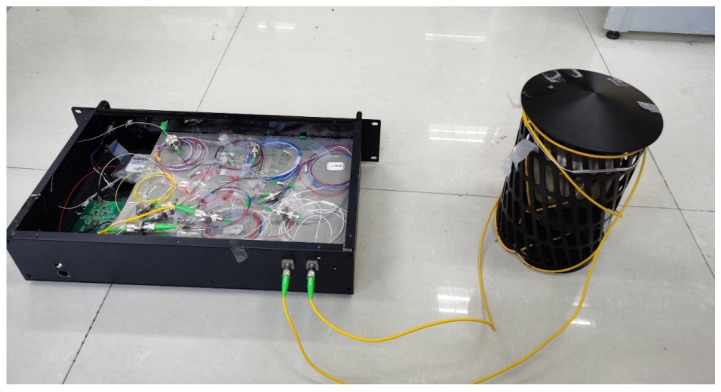
A photo of the testing system.

**Figure 7 sensors-22-08193-f007:**
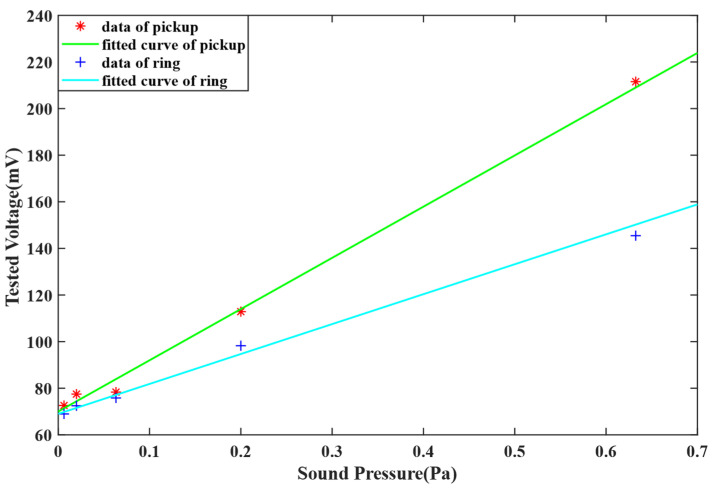
The experimental results of 500 Hz.

**Figure 8 sensors-22-08193-f008:**
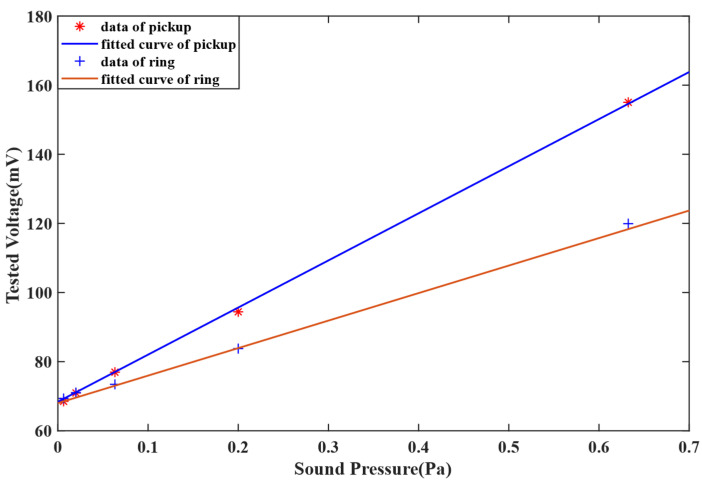
The experimental results of 1000 Hz.

**Figure 9 sensors-22-08193-f009:**
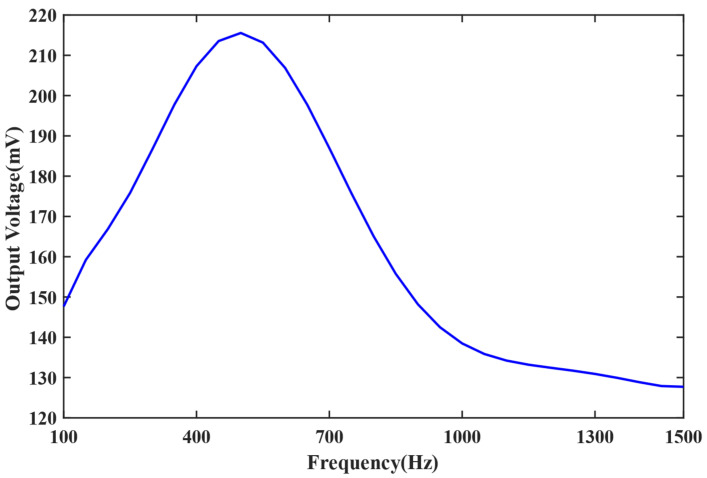
The sound frequency spectral between 100 and 1500 Hz.

## Data Availability

Data not available due to legal restrictions.

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
