# Peer review of "An Improved Acoustic Pick-Up for Straight Line-Type Sagnac Fiber Optic Acoustic Sensing System"

_sensors, 2022, doi:10.3390/s22218193_

Round 1

Reviewer 1 Report

The paper is about the improvement of a widely used acoustic sensing system, thus, the results of the research is useful and effective. The fact, that the manuscript contains both theoretical calculations and experimental results should be highlighted as a positive. I recommend the manuscript for publication after minor modifications to improve the interpretation of the study for the better understanding:

1.       The abstract is too short, it should be extended if the formal requirements of the journal allow a little bit longer abstract. The authors should write more about the theoretical analysis. What was investigated and how can we conclude the better sensitivity according to the theory. Delete the word „obviously” from line 17. If the result of the theoretical analysis is obvious, it would mean, that the work is useless, but this is not the case. Write more about how the experiments were carried out.

2.       The Sagnac system is well-known for a certain group of people, but other people would like to understand this system. Explain the working principle of this system, or at least give 1-2 references, from which the reader can get familiar with this better. Also, explain better Fig. 1: what is SLD, what are ports 2 and 4 of C1, …

3.       In equation 6 (and later), notations C1, C2 and C3 are used for different purpose than in Fig 1. This can be confusing. Check the manuscript for further matching notations.

4.       As a result of the theoretical calculations, the authors conclude, that the pick-up structure increases the phase change considerably. Is there any theoretical estimation for the increment in the sensitivity based on the increased phase change? This could connect the theoretical and experimental parts better.

5.       In Fig 5., it would be more informative to show the pick-up without the protection cover.

6.       For ultrasonic applications, the 500 Hz and 1000 Hz testing frequencies are seeming to be too low. What would be the sensitivity at higher frequencies (10 - 100 kHz or higher)?

Reviewer 2 Report

In this paper, an improved acoustic pick-up is presented to enhance the acoustic sensing sensitivity of straight-line type Sagnac fiber optic acoustic sensing system. A hollow elastomer cylinder is introduced into the system, and the optical fiber is tightly wound the cylinder to construct the pick-up. This is a rather interesting study, but I need to give a number of minor and major remarks: 

- There are a lot of fiber optic and other acoustic sensors types. The authors did not explain in details the advantages of Sagnac sensors versus others, including distributed ones (the latest refs are 10.1109/JLT.2021.3059771 , 10.3390/s22031033 , 10.3390/photonics9050277). What is the reason for their use in such case?

- Tables 1 and 2. Are these data important for a reaser? I think the approximate values can be seen on the Figs... (just as a proposal)

- The setup is not explained in details. The experiment should be repeatable. This means that the elements of the setup need to be arranged as a list or table, where their models and general params are given. The following characteristics are most important for a reader, I think: SLD model (wavelength, bandwidth, radiation power), fiber types and their params, detector, DAQ parameters, etc...

- What is the reasion for testing the pick-up only at 500 Hz and 1000 Hz? By sweeping within a frequency range you could obtain the spectral responce (an important thing for every acoustic sensor) and to compare this with other results. Again, these two frequencies need to be explained. Without the frequency responce data it should be named 'acoustic pressure sensor' or something like this.

- You talk about the comparison between two modifications of your sensing elements. And what is the place of these results among other similar state-of-the-art studies? Such a comparison needs to be added to the Discussion section, this is a very important thing.

- This is clearly seen that the measurement points from figures 7 and 8 do not strictly match the fitting curve. Is it possible to evaluate the sound preasure measurement accuracy?

- The Conclusions section looks formal. I propose to give more technical details there.

- In the reviewer account, I can see that the paper type is Communicatio, but it is written in the manuscript that this is an a Article... I propose to classify the paper as an Article, because it is rather big and should be also updated, by my opinion. But this is just a proposal...

However, by my opinion, this is an interesting and rather important study and I do hope that it will be published in Sensors as soon as all the revisions done.

Round 2

Reviewer 2 Report

Thank you so much for your work, now it looks good.I believe it can be published in Sensors in present form.

Wish you good luck with further research.